# Potential Use of Marine Seaweeds as Prebiotics: A Review

**DOI:** 10.3390/molecules25041004

**Published:** 2020-02-24

**Authors:** Aroa Lopez-Santamarina, Jose Manuel Miranda, Alicia del Carmen Mondragon, Alexandre Lamas, Alejandra Cardelle-Cobas, Carlos Manuel Franco, Alberto Cepeda

**Affiliations:** Laboratorio de Higiene Inspección y Control de Alimentos, Departamento de Química Analítica, Nutrición y Bromatología, Universidade de Santiago de Compostela, 27002 Lugo, Spain; a_lo_san@hotmail.com (A.L.-S.); aliciamondragon@yahoo.com (A.d.C.M.); alexandre.lamas@usc.es (A.L.); alejandra.cardelle@usc.es (A.C.-C.); carlos.franco@usc.es (C.M.F.); alberto.cepeda@usc.es (A.C.)

**Keywords:** prebiotic, polysaccharides, seaweed, Rhodophyceae, Phaeophyceae, Chlorophyceae

## Abstract

Human gut microbiota plays an important role in several metabolic processes and human diseases. Various dietary factors, including complex carbohydrates, such as polysaccharides, provide abundant nutrients and substrates for microbial metabolism in the gut, affecting the members and their functionality. Nowadays, the main sources of complex carbohydrates destined for human consumption are terrestrial plants. However, fresh water is an increasingly scarce commodity and world agricultural productivity is in a persistent decline, thus demanding the exploration of other sources of complex carbohydrates. As an interesting option, marine seaweeds show rapid growth and do not require arable land, fresh water or fertilizers. The present review offers an objective perspective of the current knowledge surrounding the impacts of seaweeds and their derived polysaccharides on the human microbiome and the profound need for more in-depth investigations into this topic. Animal experiments and in vitro colonic-simulating trials investigating the effects of seaweed ingestion on human gut microbiota are discussed.

## 1. Introduction

Marine seaweeds have been consumed whole by East Asian populations for centuries, if not millennia, appearing in traditional recipe books in many countries [1]. Additionally, their human consumption in Western countries has been increasing in the latest decades because of their association with improved human health. Some benefits of their consumption include a lower incidence of cancers, decreased blood pressure and blood sugar, and antiviral, anti-inflammatory, immunomodulatory or neuroprotective activities [1,2]. A mechanistic link proposed to explain the prevention of such diseases by seaweed consumption implicates the presence of diverse health-promoting bioactive compounds in seaweeds, including sulphated polysaccharides, polyphenols, pigments (chlorophylls, fucoxanthins, phycobilins), carotenoids, omega-3 fatty acids or mycosporine-like amino acids [1,3,4]. In some cases, these compounds are not produced by terrestrial plants or are not consumed in adequate quantities as part of a typical Western diet.

Nowadays, with a continuously expanding world population, fresh water is an increasingly scarce commodity, and the world’s desertification processes continue. About 45% of the world’s land surface is considered drylands, while 12 million hectares of land are degraded yearly through lack of water and related processes [5]. According to the Food and Agriculture Organization of the United Nations [6], agricultural productivity is persistently declining at over 1% per year. It is thus reasonable to expect that in the next decades there will be a need for increasing algae production to replace or supplement the intake of plant foods of terrestrial origin. Seaweeds have numerous advantages over terrestrial plants, such as rapid growth rates, and they do not require arable land, fresh water or contaminating fertilizers [7]. Additionally, an increase in seaweed cultivation would provide environmental services as an added benefit [5].

Marine seaweeds constitute approximately 25,000–30,000 different species, with a great diversity of forms and sizes [8]. The taxonomic groups, which reflect their pigmentation, include red algae (Rhodophyceae), brown algae (Phaeophyceae) and green algae [8]. Although they are still produced on a very modest scale relative to global food production, their worldwide cultivation has increased rapidly in the last decades, reaching a current yearly production of about 29 million tonnes [8]. Asian countries dominate world production, with 99% of the total, while most other maritime countries produce little or none [5]. Polysaccharides account for the majority of seaweed biomass (up to 76% of dry weight in some species; [8] and together with oligosaccharides have been the key focus of many studies of seaweed-derived compounds. Phenolic compounds and proteins from seaweeds have also attracted interest as potential functional ingredients [8]. Besides their consumption as an entire food, seaweeds or their polysaccharides are considered valuable additives in the food industry because of their rheological properties as gelling and thickening agents [2]. Additionally, seaweeds were largely employed in the formulation of animal feed [9], as well as in the formulation of cosmetics, drugs and fertilizers [10].

Regarding specific benefits on human health, seaweeds have demonstrated to exert preventive effects against several non-transmissible diseases such as cardiovascular diseases [11,12], antihypertensive [13], anti-obesity effects [14] and anti-diabetic effects [15,16], anti-cancer [17,18] or antioxidant activities [19].

Regarding cardiovascular diseases, there are large contributing risk factors that overlap and intertwine, contributing overall to the onset and growth of the disease [11]. Between them, it can be cited a cascade of mechanisms including vascular inflammation, oxidative stress, hypercoagulability and activation of the sympathetic and renin-angiotensin systems. Although in some cases the exact mechanisms through seaweeds can prevent cardiovascular diseases are not always fully understood, it was demonstrated that seaweed consumption can prevent cardiovascular diseases [11,12]. Functional oligosaccharides from seaweeds are associated with a variety of biological processes linked to hypoglycaemic and hypolipidaemic activities, although the concrete mechanisms have not been well studied [15]. With respect to hypertension, various compounds from seaweeds, such as protein-derived bioactive peptides and phlorotannins, can prevent hypertension by inhibition of angiotensin-I converting enzyme activity [13].

The potential anti-obesity activity derived from seaweeds consumption may involve a large variety of mechanisms and alterations in lipid metabolism, suppression of inflammation, suppression of adipocyte differentiation and delay in gastric emptying [14]. Between then, an important anti-obesity activity of seaweeds is the inhibition of peroxisome proliferator-activated receptor γ (PPARγ) expression and activation of the adenosine monophosphate-activated protein kinase (AMPK) phosphorylation [13]. Other important anti-obesity mechanism of seaweeds is related with inhibition of lipases, especially pancreatic lipase, that is one of the main therapeutic targets of anti-obesity drugs [14] and was recently demonstrated for various seaweeds species [20]. Additionally, anti-obesity mechanisms related with concrete seaweed components, such as phlorotannins, that target the inhibition of adipocyte differentiation or fucosterol, that decreases the expression of the adipocyte marker proteins PPARγ and CCAAT/enhancer-binding protein alpha were reported. Seaweeds can also prevent obesity by means of the modification of the relative quantities of phyla in the gut microbiota (GM), and polysaccharides from seaweeds can reduce obesity also by repairing the intestinal barrier and reducing inflammation [21].

Although abundant signaling pathways have been found to be involved in the process of glucose metabolism and anti-diabetic effects [15,16], among them, the IRS1/PI3K/JNK/AKT/GLUT4 pathways are important mechanisms in insulin signal transduction. Some seaweed components were shown in animal models to ameliorate the hepatic insulin resistance by regulating the cited signaling pathways [15,16,22]. In addition, seaweeds significantly increased the abundance and diversity of gut microbiota in animal models and showed the ability to increase the population of beneficial microbiota and maintain the homeostasis of the GM [15,22].

With respect to the antioxidant activity of seaweeds, this protective affects depends mainly on phlorotannins, secondary metabolites that exert their great antioxidant activity via the scavenging of reactive oxygen species [19]. Secondary metabolites of seaweeds are also responsible for the anti-cancer activity, whose includes different mechanisms such as repairing the intestinal barrier by intensifying the expression of the tight junction proteins via increasing the phosphorylation of MAPK and ERKT/2 genes [18], and activating the caspase cascades. Other potential mechanisms are reducing the expression of cyclin-dependent kinases and matrix metalloprotease family [17] and inducing decreased levels of pro-apoptotic metabolic signals [19].

## 2. Polysaccharides from Marine Seaweeds

Indigestible dietary polysaccharides attract attention as functional food ingredients with health benefits [7]. Most carbohydrates entering the colon are fermented in the proximal colon, which is considered a saccharolytic environment. As digesta moves through the distal colon, carbohydrate availability decreases, and proteins and amino acids become the main metabolic energy sources for bacteria in this region. The main end-products of saccharolytic fermentation are short chain fatty acids (SCFA), which contribute towards the host’s daily energy requirements. On the contrary, the end-products of proteolytic fermentation include metabolites such as phenolic compounds, and nitrogenous ones like amines and ammonia, some of which are carcinogens. This means that the GM exerts a key contribution to the human energy balance and nutrition, by extending the host metabolic capacity to indigestible polysaccharides. In addition, intestinal microorganisms contribute to develop and maintain the host immune system, defending the host from colonization by opportunistic pathogens [2]. The effects of polysaccharides on the GM are generally evaluated by the contents of SCFAs, the composition and the abundance of beneficial intestinal bacteria [20].

Polysaccharides in today’s human diet originate primarily from terrestrial plant cell walls, while other sources, such as seaweeds, are less represented [8]. Studies indicate that polysaccharides and oligosaccharides derived from seaweeds can modulate intestinal metabolism, including fermentation, inhibit pathogen adhesion and evasion, and potentially treat inflammatory bowel disease [8,23]. Some seaweed polysaccharides also demonstrate anticoagulant [24], antitumor [25], anti-inflammatory [26], antiviral, antihyperlipidemic [27] or antioxidant activity [28]. Other research has focused on their use as prebiotics to aid in limiting the occurrence of non-transmissible chronic diseases common in Western countries, such as obesity, diabetes, cardiovascular diseases or some types of cancer [2]. Nevertheless, many seaweed fibres are high-molecular-weight polymers that need to be transformed into oligosaccharides to increase their fermentability by the GM [2].

Depending on the taxonomic classification of algae, polysaccharides can vary greatly in their composition [8]. Seaweeds feature an integrated network of biopolymers in their cell walls, mainly formed by polysaccharides associated with other compounds, such as proteins, proteoglycans, polyphenols and some mineral elements, like calcium and potassium [8]. It is because of this complexity that for most seaweed polysaccharides, the exact structures, constituents and chemistry are not fully known [29]. Depending on the algal taxa, both structural and storage polysaccharides may vary. Structural polysaccharides are the most abundant, and their composition can be influenced by the seaweed species [29], as well as environmental factors, such as salinity, water temperature and sunlight intensity [30]. Some of the structural polysaccharides are carboxylated or sulphated, which can affect their fermentability [2].

Green seaweeds contain mostly sulphated structural polysaccharides, like ulvans (the most abundant, representing 8–29% of dry weight) and sulphated galactans, xylans and mannans. These polymers are composed mainly of rhamnose, xylose, glucose, glucuronic acid and sulphates, with smaller amounts of mannose, arabinose and galactose [31,32]. These polysaccharides are not fully fermented by the human GM [33,34]. Conversely, the main carbohydrate in storage is starch.

Contrariwise, brown seaweeds contain mainly cellulose, alginic acids, fucoidans and sargassans as structural polysaccharides, while the storage polysaccharides are alginates [35] (the most abundant at 17–45% of dry weight), fucoidans and laminarins [2,8,36]. Finally, red seaweeds contain agars, carrageenans, xylans, sulphated galactans and porphyrins as main structural polysaccharides, while the main storage polysaccharide is starch [21,37].

Through a long time of co-evolution between GM and host, the intestinal microbes have evolved diverse strategies for degrading polysaccharides from terrestrial plants [38]. However, because the consumption of seaweeds polysaccharides was not common over human evolution, the human GM did not acquire the same efficacy to degrade seaweed polysaccharides. Thus, although humans possess the enzymes necessaries to degrade some algal polysaccharides, such as starches, they are unable to digest the most complex polysaccharides [2,38]. An elegant work carried out by Hehemann et al. [39], showed that specific genes coding for enzymes with potential capacity to degrade seaweed polysaccharides, as porphyranases or agarases, can be transferred from a member of marine Bacteroidetes, *Zobellia galactanivorans* to the GM bacterium *Bacteroides plebeius* in Japanese individuals. As a consequence, GM of those subjects acquires the ability to degrade porphyran and agarose, as compared to the GM from North American individuals, who are incapable of degrading it [39].

De Jesus Raposo et al. [40] suggested that most seaweed polysaccharides can be regarded as dietary fibre, as they are resistant to digestion by enzymes present in the human gastrointestinal tract, reaching the distal gut. In this allocation, polysaccharides are fermented and become food for the commensal bacteria, stimulating their growth. For this reason, great efforts have been placed on developing efficient methods for seaweed polysaccharides extraction, purification and structural characteristics elucidation in order to improve their bioavailability, especially for insoluble fibre [2].

It was previously shown that variations in the chemical structure of a prebiotic can impact its selective fermentation by bacteria [41]. For this reason, there were published large works regarding the investigation of the potential prebiotic effect of single polysaccharides, often contained in seaweeds, but employing pure standards [42,43,44]. However, it should be considered that seaweeds contain other components than also can affect GM. Consequently, trials employing whole seaweeds are required to investigate their real effect in human GM. Furthermore, although several works investigated the effects of seaweeds in livestock gut microbiota, they are more oriented to study the effects of seaweeds in animal production and welfare. An elegant review was recently published where they can be checked [32].

## 3. Other Bioactive Compounds from Marine Seaweeds

In addition to polysaccharides, seaweeds also contain other bioactive compounds, called secondary metabolites much of them with antioxidant activity [45]. Among these, polyoletides (such as phlorotannins), isoprenoids (such as terpenes, carotenoids and steroids), alkaloids and shkimates (such as flavonoids) are the main groups of secondary metabolites found in algae [46]. Compared with other macroalgae, red seaweeds are richer sources of these secondary metabolites [47]. The human health benefits afforded by these bioactive compounds include anti-inflammatory, antioxidant, anticoagulant, antiviral, antimicrobial, antidiabetic, antitumor, antihypertensive, antiallergic and immunomodulatory activities [45,46,47,48].

Exceptionally, phlorotannins or polyphenols are recognized as structural classes of polyketides, found primarily in brown algae. These compounds can also reach the large intestine where GM can convert them into beneficial bioactive metabolites. Phlorotannins are highly hydrophilic components formed by polymerization of monomeric units of phloroglucinol (1,3,5-trihydroxybenzene). There are six main groups: fucols, floretoles, fucofloretols, fuhalols, isofuhalols and eckols, and all display strong antioxidant properties and act against oxidative stress [46]. Certain polyphenols are used as prophylactics against problems such as cardiovascular diseases, cancers, arthritis and autoimmune disorders [47]. In addition, some phlorotannins have been shown to decrease blood glucose levels after carbohydrate-rich meals. This action is achieved by interfering with the enzymes amylase and sucrase that intervene in the digestion and assimilation of these carbohydrates. In addition to their effects on the metabolic functions of the host, phlorotannins also appear to have some antibacterial activities [47], which may explain the low production of AGCC derived from seaweeds in which they are an important part [7]. However, like other phenolic compounds, bacterial growth inhibition occurs selectively in microbial populations, including some pathogens, and its antibacterial effect is minor in commensal bacteria [7]. Because of this selective inhibition of bacterial pathogens, large whole seaweeds and seaweeds ethanolic extracts has been used to extend the shelf life of fresh fishery foods, such as *Fucus spiralis* [48,49], *Bifurcaria bifurcata* [50], *Cytoseira compressa* [51] or *Gracilaria verrucosa* [52] Another of its potentially therapeutic functions is that extracts with high phlorotannins content have demonstrated a potent inhibitory action on the growth of cancerous cell lines [53,54].

Bromophenols present in marine algae have attracted much attention in the field of antimicrobial agents [55]. Previous studies indicate that marine bromophenols possess promising antibacterial [56,57] and antiviral activities [58]. In addition, symphyocladin G, a new bromophenol adduct derived from the red seaweed *Symphyocladia latiuscula*, is found to have antifungal activity against *Candida albicans* [59]. Several bromophenols isolated from the red alga *Odonthalia corymbifera*, are promising candidates for antifungal agents in crop protection [56]. These properties are not exclusive to red algae because compounds, such as bis(2,3-dibromo-4,5-dihydroxybenzyl) ether, isolated from brown algae *Leathesia nana*, showed cytotoxic activity against some cancer cells [54], and exhibited antibacterial activity against several strains of Gram-positive and Gram-negative bacteria [56].

With respect to terpenes, there are about 200 different diterpenoids, of which some have important cytotoxic and antiviral, antimicrobial and antiparasitic activities (such as against *Leishmania*) [60,61]. These compounds are found in red and brown algae.

Flavonoids and their glycosides are present in green, brown and red algae. These compounds possess antioxidant properties and have demonstrated action against arteriosclerosis and cancer [45]. Within this group, fucoxanthin, β-carotene and violaxanthin stand out. Besides its strong anticancer activity, fucoxanthin has promise in preventing obesity [62]. The correlation between a carotenoid-rich diet and a low risk of cardiovascular and ophthalmological diseases has been supported by recent research with different types of carotenoids in cellular systems and human intervention studies [63]. Specifically, flavonoids from *Enteromorpha prolifera* influenced the GM balance in diabetic mice, increasing the presence of *Alistipes*, *Lachnospiraceae* and *Odoribacter* genera [63]. *Alistipes* spp. is one of the most abundant bacterial genera in the mouse intestine and is capable of fermenting glucose and lactic acid to produce propionic, acetic and succinic acid, which modulate the release of intestinal hormones, thereby influencing the release of insulin and appetite. It is perhaps for this reason that *E. prolifera* is traditionally used in China as a natural herb to treat diseases associated with inflammation [63]. It has recently been observed that a polysaccharide of *E. prolifera* could be used as a novel agent to treat obesity and hyperlipidaemia [62].

Other important secondary metabolites contained in seaweeds and responsible of important beneficial effect in human health are peptides, such as lectins [48]. Lectins primarily show antiviral, antibacterial, and antifungal activities. Specially one type of lectin, griffithsin, showed important antiviral activity and is nowadays considered a promise antiviral agent, with great potential concerning the prevention of sexually transmitted infections [48], including HIV [64]. Other important peptides are renin inhibitor tridecapeptide [65] and dipeptide [66], which demonstrated hypotensive effect dipeptide. Phycoerythrin [43] and kahalalide F [67] are other important peptidic compounds isolated from seaweeds than showed antitumor effect.

Besides the so-called secondary metabolites, seaweeds contain other minor nutrients of immense importance for human health. Phycobiliproteins, responsible for the characteristic bright pink appearance of red algae, are classified into phycoerythrin (red) and phycocyanin (blue). These pigments are used commercially in food, nutraceuticals, and for their therapeutic properties, mainly antimicrobial, antioxidant, anti-inflammatory, neuroprotective, hepatoprotective, immuno-modulatory and anticancer effects [67,68,69,70,71,72]. Such compounds may improve the efficacy of standard anticancer drugs, decrease their side effects, and act as photosensitisers for the treatment of tumour cells [30].

## 4. Effects of Seaweed Polysaccharides on Human Health

Compounds with prebiotic activity, such as oligosaccharides, lactulose, fructo-oligosaccharides (FOS), inulin, galacto-oligosaccharides and arabinoxylano-saccharides are used as functional ingredients in the food industry [10]. While most of the above compounds are now derived from terrestrial plants, some studies have shown that polysaccharides and oligosaccharides derived from marine algae can also modulate intestinal metabolism, including fermentation, inhibit adhesion and invasion of pathogens, and treat inflammatory bowel disease [23,73]. Furthermore, these compounds have demonstrated anticoagulant, antioxidant, immunomodulatory, antitumor and antiviral activities [10].

Being much less degradable by enzymes from the human upper gastrointestinal tract than their terrestrial plant counterparts, polysaccharides from marine algae reach a greater proportion in the descending colon. For this reason, some authors [42,74] have found that polysaccharides from marine algae, such as alginate, agarose oligosaccharides and κ-carrageenan oligosaccharides, have a higher prebiotic activity than FOS in vitro. Specifically, sulphated polysaccharides from marine algae show anticoagulant, antiviral, antitumor, anti-inflammatory, antibacterial, immunological, antioxidant and many other biological and physiological activities [8,75]. Sulphated polysaccharides include fucoidans (l-fucose and sulphated ester groups) from brown seaweeds, agars and carrageenans (sulphated galactans) from red seaweeds, and ulvans (sulphated glucuronoxylorhamnan) and other sulphated glycans from green seaweeds [8].

The consumption of these sulphated polysaccharides can block the adhesion of leukocytes to the epithelium of blood vessels, preventing the migration of these cells to the site of inflammation [76]. These polysaccharides often stimulate the growth and activity of beneficial bacteria by acting as substrates for fermentation in the large intestine, leading to the production of SCFA, with multiple functions that help maintain health [8]. As previously mentioned, seaweed polysaccharides differ in their properties and compositions from one type of algae to another [46], so their effects on the human GM will also differ.

The GM, especially in its most distal parts, harbors many bacteria, archaea, protozoa and viruses, which along with their genetic material, is collectively referred to as the gut microbiome (GMB) [77]. This GM is composed of up to 12 different bacterial phyla of which more than 90% belong to the Proteobacteria, Firmicutes, Actinobacteria and Bacteroidetes [78], while the remaining phyla are much less constant and numerous [79]. The most frequent bacterial species in the colon, which is where the highest bacterial concentration exists [78], belong mainly to the families *Bacteroidaceae*, *Prevotellaceae*, *Rikenellaceae*, *Lachnospiraceae* and *Ruminococcaceae* [77]. The GM presents a diverse set of functions important to human health, such as the extraction of energy from a broad spectrum of nutrients, the production of vitamins, the promotion of immune homeostasis and the prevention of colonization of the intestine by pathogens [80]. One of the most important functions of the GM is in the prevention of chronic low-grade inflammation [81]. Host genetics define the chemistry and physics of the GM, including the availability of nutrients and the threshold of activity required to induce an immune response. Consequently, intestinal microbial communities are composed of species that have evolved to occupy specific ecological niches in the gut, including the ability to metabolize specific molecules available from the host or to evade host defenses [77].

Nutrients can interact directly with the GM to promote or inhibit its growth. In this sense, the ability of the GM to extract energy from specific components of the diet offers a direct competitive advantage to specific members of the GM, allowing them to proliferate at the expense of other members [81]. Thus, diet affects not only the composition and absolute abundance of intestinal bacteria but also their growth kinetics [82]. In this context, the most influential nutrients are indigestible carbohydrates, which can be of both terrestrial and marine algae origin [81].

The human genome encodes a limited number of hydrolases capable of hydrolyzing the glycosidic bonds of polysaccharides in dietary fibre (collectively referred to as CAZymes). Consequently, many polysaccharides, such as resistant starch, inulin, lignin, pectin, cellulose and FOS, reach the large intestine undigested. In contrast, the GMB codes tens of thousands of CAZymes. In the presence of bacteria harboring key enzymes involved in carbohydrate metabolism, these complex polysaccharides can thus be degraded and metabolized in vivo [83]. The bacteria able to degrade these complex polysaccharides are called primary degraders and include members of the genera *Bacteroides*, *Bifidobacterium* and *Ruminococcus*, *Roseburia*, *Facealibacterium*, *Anaerostides* or *Coprococcus*. A relative abundance of these genera in our GM infers that during a food shortage, these bacteria can alternate between energy sources by using sensors and regulatory mechanisms that control gene expression [21,81]. Hydrolases act on polysaccharides to generate oligosaccharides and monosaccharides. Secondary fermentation of these compounds by the GM produces SCFA, specifically acetic, propionic, butyric, lactic and succinic acids, which initiate a complex metabolic network [81].

The GM of hunter–gatherer, rural and agricultural populations are usually more bacterially diverse than in modernized urban societies [84] and so require a greater functional repertoire to maximize their energy intake from dietary fibres. Conversely, the consumption of a diet composed mainly of products of animal origin causes an enrichment in the GM of genera of bile-tolerant bacteria, such as *Alistipes*, *Bilophila* and *Bacteroides*, and the almost total exhaustion of bacteria that metabolize polysaccharides, such as *Roseburia*, *Eubacterium rectale* subgroup and *Ruminococcus bromii* [81].

Clinical studies investigating prebiotic effects have some disadvantages with respect to ethical constraints, as well as limited sampling possibilities from the colon and limited measurements of in situ SCFA production. These concerns are commonly avoided by applying an in vivo approach [41], that are the most common in the investigation of seaweed effects on human GM, as is described below. Contrariwise, in vitro studies show important limitations because only represent the first step of a long process, and the results observed in vitro can be magnified, diminished, or totally different in a more complex and integrated system [48]. An additional limitation is that, due to the short fermentation time in in vitro studies, they fails to capture the complete picture of cross-feeding interactions between gut microbes, and which may not fully correlate with the long-term effects of seaweed compounds on GM [41].

### 4.1. Polysaccharides from Brown Seaweeds

Although the prebiotic and immuno-modulation properties of brown algae have been studied both in animal models and in vitro, humans intervention studies are also needed to assess whether there is a direct association between these uses of algae and the human GM, but are currently restricted due to ethical concerns [78]. The most relevant results obtained from examining the impacts of brown seaweeds on the GM can be found in Table 1. In this table it were included results about the prebiotic effect of brown seaweed species from genus *Ecklonia* [7,85], *Sargassum* [86,87,88], *Laminaria* [82,89,90,91,92], *Ascophyllum* [93,94,95], *Fucus* [23,63], *Undaria* [90], *Saccorhiza* [96] or *Porphyra* [97]. As can be seem in Table 1, in most cases, the administration of whole brown seaweed or brown seaweed-extracted polysaccharides resulted in an increase of SCFA production, stimulating of beneficial bacteria grown such as *Lactobacillus* [7,82,85,86,95], *Bifidobacterium* [7,82,85,87,92] or *Faecalibacterium* [7,58,87]. In some cases, the brown seaweed or brown seaweed-extracted polysaccharides also inhibited the growth of potentially pathogen bacteria [73,86]. In some cases, it were reported other beneficial effects not strictly related with action on GM, such as reducing serum inflammatory markers [23], reducing serum levels of lipopolysaccharide-binding protein [44], increasing CAZymes [44], reducing activity of fecal bile salt hydrolase activity [96], or reduced the expression or diabetes-related genes [15].

Laminar storage polysaccharides, typical of brown seaweeds, are low-molecular-weight, linear polysaccharides composed of glucose units with a low degree of branching [79]. Besides affecting mucin composition and SCFA concentration, laminins can affect the adherence, translocation and proliferation of bacteria in the gut [98,99]. At the same time, laminins stimulate the proportion of *Bifidobacterium*, which generates a prebiotic potential. In other research, laminarin has been shown to promote an immune response [98], and could be useful for inhibiting the production of putrefactive substances from undigested proteins [100]. In vitro batch fermentation of laminarin for 24 h promoted an increase in *Bifidobacterium* and *Bacteroides*, and propionate and butyrate production [42]. Contradicting results by other researchers indicated that laminarin was not selectively fermented by *Lactobacillus* and *Bifidobacterium*, but could modify the composition, secretion and metabolism of the jejunal, ileal, caecal and colonic mucosa to protect against bacterial translocation [32]. In addition, laminarin increased the presence of *Clostridium* spp. and *Parabacteroides distasonis* in rats [101].

An in vitro study conducted with the species *Sargassum thunbergii* revealed a dramatic increase in the population of beneficial bacteria (from 17% to 28%), while a group of harmful Firmicutes decreased from 75% to 64% after 48 h of fermentation [87]. No noticeable changes were found in Proteobacteria or Actinobacteria. At the genus level, an increase in *Lactobacillus*, *Bifidobacterium*, *Roseburia*, *Parasutterella* and *Fusicatenibacter* appeared after incubation for 24 h, followed by an increase in *Faecalibacterium* and *Coprococcus* at 48 h of incubation [87]. *Bifidobacterium*, *Coprococcus* and *Parasutterella* have been negatively correlated with non-alcoholic steatohepatitis, hepatocellular carcinoma and diabetes [102], while *Ruminococcus*, *Roseburia* and *Faecalibacterium* are producers of butyric acid and are facilitate the degradation of polysaccharides and fibres [103]. *Fusicatenibacter* was positively associated with increased serum leptin in obese rats [104], which reduces their appetite. All these findings highlight the prebiotic potential of *S. thunbergii* by its modulation of the composition and abundance of beneficial GM.

An in vitro study using *S. wightii* in MRS broth evaluated their antioxidant activity and prebiotic score comparing *L. plantarum* and *Salmonella* Typhimurium relative growths. The study showed that the prebiotic activity score was positive, promoting selectively the growth of *L. plantarum* with respect to the pathogen *S. Typhimurium*. Specifically, a prebiotic effect by 1.42-fold more growth stimulation of *L. plantarum* than *S.* Typhymurium [86].

In other work Chen et al. [58] showed an increase in fucoidan from *A. nodosum* in an in vitro assay simulating the human digestive tract was due to an increase in Bacteroidetes, Firmicutes and SCFA. At the genus level, the genera *Bacteroides*, *Phascolarctobacterium*, *Oscillospira* and *Faecalibacterium* increased, while the levels of *Fusobacterium*, *Megamonas*, *Parabacteroides*, *Clostridium* and *Dorea* decreased relative to the samples to which the algae *A. nodosum* had not been added [46]. demonstrated the in vitro prebiotic activity of a mixture of fucoidans and alginates obtained from *A. nodosum*, leading to an increase in the growth rate of *L. delbrueckii* and *L. casei* to levels similar to those observed after administration of inulin, a standard commercial prebiotic [46]. Other authors [93] conducted a study in rats, which were administered polysaccharides extracted from *A. nodosum*, and they were seen an increase in both acetate, propionate and butyrate SCFAs.

According to Zaporozhets et al. [76], fucoidans obtained from *F. evanescens* stimulate the colonic growth of beneficial *Bifidobacterium* species, such as *B. longum* B379M and *B. bifidum* 791B. Lean et al. [23] administered *F. vesiculosus*-derived fucoidan extracts to mice and, interestingly, found a reduction in markers associated with inflammatory bowel diseases.

When Wister rats were fed with feed enriched with alginates or laminarins, An et al. [101] found a notable decrease in the number of metabolites resulting from putrefaction, such as indole, H_2_S and phenol. This result was subsequently confirmed in both in vitro and rat models by Nakata et al. [100], who also found a decrease in ammonium levels with alginate. At the phyla level, alginate increased the levels of Actinobacteria, while laminarins increased the levels of Proteobacteria. At the genus level, *Bacteroides* was markedly more abundant in the group fed with alginate, and *B. capillosus* was the most frequent species. In rats fed with laminarin-enriched feed, *Parabacteroides*, *Lachnospiraceae* and *Parasutterella* bacterium were detected in greater abundance than in control rats. Nguyen et al. [44] studied laminarin supplementation in a mice high-fat diet. They could see a decrease in Firmicutes and an increase in the Bacteroidetes phylum, especially the genus *Bacteroides*.

Ramnani et al. [94] performed in vitro fermentation with *A. nodosum*-derived alginates, which increased *Bifidobacterium* and SCFAs. An increase in the proportion of Bacteroidetes to Firmicutes was observed as well in fermentations added with sulphated polysaccharides extracted from *A. nodosum* versus controls. Increased levels of Bacteroidetes and decreased levels of Firmicutes have been associated with a reduced risk of obesity in humans [79].

Evidence that probiotic bacteria in the gastrointestinal tract utilize dietary alginate was reviewed by Shang et al. [38]. Among the studies, Kuda et al. [73] found that supplementation with sodium alginate and laminarin of brown algae inhibited the adhesion and invasion of pathogens, such as *S. Typhimurium*, *Listeria monocytogenes* or *Vibrio parahaemolyticus*. Other authors reported an increase in *Lactobacillus* and *Ruminococcus* in the intestine of mice fed fucoidans from *A. nodosum*, besides a reduction in the opportunistic *Peptococcus* bacteria [95].

In vitro fermentation experiments conducted by Charoensiddhi et al. [85] demonstrated the growth-promoting effect of *E. radiata* extracts on beneficial bacteria, such as *Bifidobacterium*, *Lactobacillus* and *Clostridium coccoides*, and SCFAs production was stimulated as well. Later, the same authors [7] found increased levels of beneficial bacteria, such as *Bifidobacterium*, *Lactobacillus* and *C. coccoides* associated with the phlorotannin-enriched fermentation of *E. radiata*. Higher numbers of *Lactobacillus*, *Faecalibacterium prausnitzii*, *C. coccoides*, Firmicutes and *E. coli* were observed for phlorotannin-supplemented fermentation compared with inulin fermentation [7]. In contrast, the number of *Enterococcus* in both fermentations decreased approximately ten-fold relative to the initial counts.

Other authors tested the effects of supplementation of two brown algae (*U. pinnatifida* and *L. japonica*) on the GM and body status of laboratory rats [90]. In both instances, the animals’ body weight was reduced, which was thought to be mediated by the influence of the seaweed on the composition of the intestinal microbial communities associated with obesity, reducing the proportion of Firmicutes with respect to Bacteroidetes, and the populations of pathogenic bacteria, such as *Clostridium*, *Escherichia* and *Enterobacter* [90]. Similarly, *L. japonica* increased beneficial bacteria and SCFA, and decreased the pH level [82], while β-glucans extracted from *L. digitata* increased *Bifidobacterium* and propionic and butyric acids in vitro, in addition to lowering pH [92].

β-Glucans obtained from another *Laminaria* species (*L. digitata*) were able in an in vitro test [92] to increase *Bifidobacterium* and propionic and butyric acids, in addition to lowering pH. A study by Strain et al. [91] in vitro investigated the effect of a polysaccharide-rich raw extract obtained from *L. digitata*. A significant alteration of the relative abundance of several families, including *Lachnospiraceae* and genera such as *Streptococcus*, *Ruminococcus* and *Parabacteroides* of human faecal bacterial populations was seen. Concentrations of acetic acid, propionic acid, butyric acid and total SCFA were significantly higher.

Finally, Huebbe et al. [96] conducted a study on mice that were administered polysaccharides from *S. polyschides* with a high-fat diet. A metabolic improvement was seen including normalization of blood glucose, reduction of plasma leptin, reduction of fecal bile salt hydrolase activity and secondary bile acids in these mice.

### 4.2. Polysaccharides from Red Seaweeds

The most relevant results obtained from the investigation of red seaweeds effect on GM can be found in Table 2. In this table, results about the prebiotic effect of red seaweed species from genus *Acanthopora* [86], *Gracilaria* [105,106], *Kappaphycus* [107], *Euchema* and *Grateloupia* [10], *Chondrus* [57], *Gelidium* [94], or *Osmundea* [88] were included.

As can be seem in Table 2, as was described previously for brown seaweeds, administration of red seaweeds or seaweed-extracted polysaccharides resulted in an increase of SCFA production, stimulating of beneficial bacteria grown such as *Lactobacillus* [86] or *Bifidobacterium* [57,94,107], whereas inhibited the growth of potentially pathogen bacteria [57,86]. It was also reported red seaweeds activity on the prevention of naproxen-induced gastrointestinal damage [106].

Agarose stands out among the polysaccharides isolated from red algae that cannot be digested by human intestinal enzymes. When seaweed is consumed, whether as an edible food or food additive, agarose reaches the most distal portions of the gastrointestinal tract, where it is fermented and metabolized by the GM [108,109]. As described by Ramnani et al. [94], low-molecular-weight agarose exerted a prebiotic effect in vitro by promoting the growth of *Bifidobacterium* and increasing SCFA concentrations in the medium.

Bajury et al. [107] conducted an in vitro colon model in which they evaluated the prebiotic capacity of *K. alvarezii*. This study showed an increase in SCFA (particularly acetate and propionate) and *Bifidobacterium*. In the other hand, decrease in *C. coccoides* and *E. rectale.* These results suggested that *K. alvarezii* might have the potential as a prebiotic ingredient. A study published by Zhang et al. [110] focused on the beneficial effect of low-melting-point agarose (in the form of neoagaro-oligosaccharides) on the GM during the relief of intense exercise-induced fatigue in mice. Results showed the abundance of Bacteroidetes and Proteobacteria increased and decreased, respectively, during the attenuation of fatigue and its associated gastrointestinal problems. Ladirat et al. [111] found that mice fed 2.5% (*w*/*v*) neoagaro-oligosaccharides for seven consecutive days achieved a much more pronounced increase in the population of *Lactobacillus* spp. and *Bifidobacterium* spp. in their GM relative to those fed 5% (*w*/*v*) FOS for 14 consecutive days. Likewise, it was demonstrated that agaro-oligosaccharides could be used as a prebiotic to encourage the growth of beneficial strains of bacteria, such as *B. adolescentis* ATCC 15703 and *B. infantis* ATCC 15697. Low-molecular-weight agar has demonstrated a bifidogenic effect, along with an increase in SCFA acetate and propionate concentrations, after 24 h of in vitro fermentation with human faeces inoculant [94].

Another type of polysaccharide with prebiotic function found in red algae are the group of carrageenans, which are derived from D-galactose, and approved as food additives [79]. In rats fed 2.5% *C. crispus*, of which carrageenan is a major polysaccharide, *B. brevis*, as well as SCFA, increased considerably, while pathogens *Clostridium septicum* and *Streptococcus pneumonia* noticeably decreased compared with the basal diet [57]. Elevation of plasma immunoglobulin levels was also found in rats fed with *C. crispus*, resulting in improved host immunity. Consistent with the prebiotic activity of carrageenan, carrageenans isolated from red algae *G. filicina* and *E- spinosum* promoted the growth of *Bifidobacterium* [10].

Research led by Di et al. [105] found that the polysaccharides of *Gracilaria rubra* increased the relative abundances of *Bacteroides*, *Prevotella* and *Phascolarctobacterium* in vitro compared with the control group. *Bacteroides* spp. assists the host with degrading polysaccharides and contains codifying genes of glucosidase enzymes [39]. *Prevotella* is another beneficial genus with the potential to participate in the metabolism and utilization of plant polysaccharides. The genus *Phascolarctobacterium* is associated with the production of SCFA [110].

Many other bacterial genera, such as *Legionella*, *Sutterella*, *Blautia*, *Holdemania*, *Shewanella* and *Agarivorans*, were decreased as a consequence of intake of *C. crispus* supplements in rats [57]. Decreases in the presence of *Streptococcus* were also observed. In conclusion, carrageenans from *C. crispus* could act as a fermentable substrate for probiotic bacteria present in the gastrointestinal tract, thereby promoting the growth of probiotic groups, while inhibiting certain groups of pathogenic bacteria [57]. Another study in chickens described an overall impact of administering whole red algae (*Sarcodiotheca gaudichaudii* and *C. crispus*) on the intestinal mucosa, increasing the height and surface of the villi in these animals [112]. Moreover, the abundance of beneficial bacteria, such as *B. longum* and *Streptococcus salivarius* increased, while some harmful bacteria species, such as *C. perfringens*, decreased [112] Rodrigues et al. [88] used extracts from the red algae *O. pinnatifida* and *S. muticum* in an in vitro fermentation system, which increased the production of acetate and propionate, and the population of *Bifidobacterium*. In work published by Silva et al. [106], in which extracts of sulphated polysaccharides from *G. birdiae* were administered to laboratory rats, gastrointestinal damage induced by naproxen was prevented, although it did not produce notable variations in the GM of these rats.

An in vitro study using *A. spicifera* in MRS broth evaluated their antioxidant activity and prebiotic score with *L. plantarum* and *S. Typhimurium*. The study showed that the prebiotic activity score was positive, promoting the growth of *L. plantarum* and suppressing the growth of the pathogen *S. Typhimurium*. Specifically, a prebiotic effect by 0.84-fold more growth stimulation of *L. plantarum* than *S. Typhymurium* [86].

### 4.3. Polysaccharides from Green Seaweeds

Unlike brown and red algae, the current evidence for the fermentation capacity of green algae and their polysaccharides is scarce, partly because their fermentation requires a specific activity of α-l-rhamnosidase in the gastrointestinal tract, which is infrequent [113]. The most relevant results obtained from the investigation of green seaweeds effect on GM can be found in Table 3. In this table, results about the prebiotic effect of green seaweed species from genus *Enteromorpha* [38,82,86,113,114,115] and *Ulva* [115] were included. Administration of green seaweeds or seaweed-extracted polysaccharides also resulted in an increase of SCFA production, stimulating of beneficial bacteria grown such as *Lactobacillus* [38,86,115], *Bifidobacterium* [38], or *Akkermansia* [38] whereas inhibited the growth of potentially pathogen bacteria [81,114]. Other beneficial actions were reported such as decrease lipopolysaccharide-binding protein in female mice [38], diminished histopathological lesions of inflammatory infiltrations in distal colon [114], or modulating diabetes-related genes expression in diabetic mice [22].

Ulvans are one of the most frequent polysaccharides in green algae. This polysaccharide is a water-soluble sulphated heteropolysaccharide [79]. Ulvans contains sulphate and uronic acids, and so produce undigestible ionic colloids, has ion-exchange capacity and can bind to bile acids, consequently increasing the excretion of bile acids with cholesterol-lowering or antihyperlipidemic effects [2,79]. Antioxidant and immunomodulatory properties are other beneficial actions elicited by ulvans [116,117].

Ulvans has also been studied for its possible prebiotic potential. Kong et al. [82] performed an in vitro assay using *Enteromorpha* with a high content of ulvans, but there were no noticeable variations in the populations of *Enterococcus*, *Lactobacillus* and *Bifidobacterium* compared with controls. In contrast, in a recent in vitro faecal fermentation analysis, ulvans stimulated the growth of *Bifidobacterium* and *Lactobacillus* populations and promoted the production of SCFA, such as lactic and acetic acids [42]. In a previous study by Ren et al. [114], both whole *Enteromorpha* and polysaccharides extracted from *Enteromorpha* improved inflammation associated with loperamide-induced constipation in mice. In those mice, the GM showed an increase in Firmicutes and Actinobacteria compared with the control mice, whereas the relative amounts of Bacteroidetes and Proteobacteria decreased.

In work by Shang et al. [38], an extract of *E. clathrata* was administered to mice, resulting in marked decreased concentrations of genera, such as *Enterobacter*, *Staphylococcus* and *Streptococcus*. Surprisingly, such supplementation also dramatically reduced the population of *A. muciniphila* in the intestine. These observations indicate a possible unfavorable effect of these polysaccharides on the GM. Contrarily, these polysaccharides were reported to increase the abundance of *A. muciniphila*, *Bacteroides*, *Alloprevotella*, *Ruminococcaceae* and *Blautia* in the intestinal tract of mice, and decrease the abundance of *Peptococcus*, *Rikenellaceae* and *Alistipes* [96,109].

An in vitro faecal fermentation of xylans derived from *Palmaria palmata* reported that xylose was fermented after 6 h, and the SCFA content increased simultaneously [32]. This study did not determine the bacterial composition. Nonetheless, Xylans and xylo-oligosaccharides extracted from terrestrial plants, such as wheat husks and corn, are considered potential prebiotics due to evidence of bifidogenesis, improved plasma lipid profile and positive modulation of immune function markers in healthy adults [118]. 

*E. clathrata* is an edible green seaweed possessing polysaccharides with numerous bioactivities, including anticoagulant, immunomodulatory, antioxidant, anticancer and anti-obesity effects [38]. It was reported that the polysaccharides of *E. clathrata* exerted diverse prebiotic effects on *A. muciniphila*, *Bifidobacterium* and *Lactobacillus* in male and female mice [38]. The results were most evident in the male mice because of a sex-specific effect on the GM, as sex hormones play a key role in determining the composition of intestinal microorganisms [109]. In other work from the same authors, male mice were supplemented with polysaccharides of *E. clathrata* in the diet, which increased the abundance of *Bacteroides*, *Prevotella*, *Alloprevotella*, *Eubacterium* and *Peptococcus*, and decreased the proportion of the cancer-related *Helicobacter* [109]. In the female counterparts, the abundance of *Odoribacter*, *Clostridium* IV, *Oscillibacter* and *Alistipes* spp. increased, and the proportions of beta-proteobacteria decreased [38].

An in vitro study using *E. compressa* in MRS broth evaluated their antioxidant activity and prebiotic score with *L. plantarum* and *S. Typhimurium*. The study showed that the prebiotic activity score was positive, promoting the growth of *L. plantarum* and suppressing the growth of the pathogen *S. Typhimurium*. This seaweed exhibited the highest score of prebiotic activity (1.44-fold), stimulating the growth of *L. plantarum* than *S. typhimurium* [86].

The natural products of marine macroalgae have shown notable antidiabetic potential by interfering with carbohydrate metabolism. For example, *E. prolifera* contains many bioactive compounds, such as sulphated polysaccharides, which could improve glucose metabolism, in addition to displaying anti-inflammatory, antiviral and anticoagulant functions [22].

## 5. Conclusions

Although substantial evidence of the prebiotic effect of seaweed and seaweed extracts has been published in recent years, these studies have been performed using in vitro digestion systems simulating the human colon, or in animal models. Animals, such as mice or rats, differ widely from humans in the GM composition, immune function, diets, metabolism and other key aspects, so extrapolating the results obtained from animal models to humans may not be valid. In vitro systems replicate more similarly the human intestinal microbiota, but are less-dynamic systems than the real human colonic environment. Additionally, other factors should be considered, such as the possible effect of other secondary compounds contained in seaweeds on the GM composition, or the potential to transfer genes from marine bacteria to human GM bacteria coding for enzymes that could degrade seaweed polysaccharides. Thus, not all people will respond equally after seaweed ingestion. The decrease in terrestrial agriculture and disposable water is likely to increase the consumption of algae by humans in the near future. Meanwhile, there is a profound need for more in-depth investigations into the potential prebiotic effects of marine seaweeds and their derived polysaccharides on the human GM.

## Figures and Tables

**Table 1 molecules-25-01004-t001:** Prebiotic effect of different species of brown seaweed.

Type of Study	Seaweed, Dosage and Time of Exposure	Polysaccharides Characterization	Significant Changes in Gut Microbiota	Significant Changes in Related Metabolites	Reference
In vitro fermentation system using fresh fecal samples from four healthy donors	Polysaccharides extracted from 20 g of *Ascophyllum nodosum* in a single dose, compared to blank and FOS-added samples	Total carbohydrate 42.3%; uronic acid 11%; protein 1.4% and sulfate content 23.9%; Monosaccharides content were composed of Man, GlcA, Glc, Gal, Xyl, and Fuc at a molar ratio of 16.65, 20.34, 1.60, 9.69, 3.44, and 48.29	Increase in Bacteroidetes and Firmicutes. At genus level, increase of *Bacteroides*, *Phascolarctobacterium*, *Oscillospira*, *Faecalibacterium*, while decreased *Fusobacterium*, *Megamonas*, *Parabacteroides*, *Clostridium*, *Dorea*	Increase in SCFA, acetate and propionate in *A. nodosum* added polysaccharides with respect to blank samples and FOS-added samples	[58]
In vivo trials using 3 Wistar male rats per sample, comparing effect of *A. nodosum* crude polysaccharide with hydrolyzed *A. nodosum* polysaccharides. SCFA were generated by fermentation with *Lactobacillus plantarum* BCC 5493 and *Enterococcus faecalis* BCC 39,179	0.2 g of polysaccharides extracted from *A. nodosum* per rat for 4 days, comparing crude polysaccharide with crude polysaccharide hydrolysates, alginate and hydrolyzed alginate	Crude polysaccharide contained carbohydrates 22.7%, sulphate content of 17.1% and protein content 1.34%. Hydrolysates showed 25.1–26.7% carbohydrates, 25.3–25% sulphate and 1.7–1.4% protein contents	Not provided	Increase in both acetic, propionic and butyric acids, in this order. SCFA were higher in the case of polysaccharides with lower molecular weight	[93]
In vitro fermentation system using fresh fecal samples from three healthy donors	1% *w*/*v* low molecular weight polysaccharide derivatives extracted from *A. nodosum* for 24 h. Inulin was used as positive control and cellulose as negative control	Average molecular weight 31.0 and 56.0 kDa	No significant changes in GM	Increase in total SCFA, acetic and propionic acids	[94]
In vivo trial using 18 male C57BL/6 mice. 6 mice received fucoidans extracted fro *A. nodosum*, and 6 acted as blank group	100 mg/kg/day of fucoidans obtained from *A. nodosum* for 6 weeks. Control group received saline solution	21% sulfate content; 1330 KDa molecular weight; 7.3% Man, 24.1% GlcA, 1.5% Glc, 7.2% Gal, 1.3% Xyl, 58.6% Fuc	Fucoidans administration resulted in a much more diverse cecal microbiota, increase on *Lactobacillus* and *Talassospira*, whereas	Fucoidans decreased the serum levels of lipopolysaccharide-binding protein	[95]
In vitro fermentation system using fresh fecal samples from 3 healthy donors	1.5% *w*/*v* of enzyme-assisted extracted polysaccharides from *Ecklonia radiata* for 24 h. Inulin and resistant starch were used as positive controls and glucose and cellulose were used as negative controls	48.7% total fibre, 16.1% non-digestible non-starch polysaccharides, 1.3% total starch, 43% total sugar, 3.8% protein and 4.5% total phlotorannin	Increase of total bacteria, *Bifidobacterium*, *Lactobacillus*	Increase in total SCFA, acetic and propionic acids	[85]
In vitro fermentation system using fresh fecal samples from three3 healthy donors	Polysaccharides extracts obtained by microwave-intensified enzymatic process from 4.5 g of crude *E. radiata* for 24 h. Four different seaweed fractions were employed (crude extract fraction, phlorotannin-enriched fraction, low molecular weight polysaccharide-enriched fraction and high molecular weight polysaccharide-enriched fraction. Inulin was used as positive control and cellulose as negative control	Crude extract fraction: 14.4%, fibre, 5.6% non-digestible non-starch polysaccharides, 20.6% sugar, 0.2% ManA, 0.5% Man, 17.2% Glc, 0.5% Gal, 0.3% Xyl, 1.8% Fuc, 4.6% phlorotannin. Phlorotannin-enriched fraction: 3.4% fibre, 3.4% sugar, 3.4% Glc, 13.4% phlorotannin. Low molecular weight polysaccharide-enriched fraction: 0.5% fibre, 0.4% starch, 22.7% sugar, 22.7% Glc, 2.5% phlorotannin.	Increase of *Bifidobacterium*, *Lactobacillus*, *Clostridium coccoides* in all tested fractions with respect to negative controls. Low molecular weight polysaccharide-enriched fraction showed the better fermentative results, obtaining better counts that positive controls for *Lactobacillus*, *Faecalibacterium prausnitzii*, *C. coccoides* and Firmicutes	Total SCFA were higher in crude fraction than all other fractions after 24 h fermentation. All fractions except phlorotannin-enriched fraction significantly increased SCFA production with respect to negative controls	[7]
		High molecular weight polysaccharide-enriched fraction: 62.4% fibre, 22.8% non-digestible non-starch polysaccharides, 0.3% starch, 42.1% sugar, 1.9% GulA, 7.2% ManA, 2.1% Man, 1.1% GlcA, 17.1% Glc, 1.7% Gal, 1.5% Xyl, 9.4% Fuc, 1.7% phlorotannin.			
In vivo trail using 10 C57BL6 mice per group, with previously induced colitis by supplementing 3% *w*/*v* of dextran sulphate sodium in the drinking water for 8 days	Fucoidans extracted from *Fucus vesiculosus* intraperitoneally (10 mg/kg/day) or orally (10 mg/kg/day for high purity fucoidan or 400 mg/kg/day for focus-polyphenol) for 7 days	Fucus-polyphenol: 40.2% neutral carbohydrates; 21.8% sulfates; 26.2% polyphenols; 3.6% uronic acids and 203.1 kDa peak molecular weight.High purity fucoidan: 59.5% neutral carbohydrates; 26.6% sulphates; <0.5% polyphenols; 1.4% uronic acids; 61.8 kDa molecular weight.	Not provided	Both oral fucoidan reduced cytokines associated with inflammatory bowel disease such as interleukin-1α, interleukin-1β, interleukin-10, macrophrage inflammatory protein-1α, macrophrage inflammatory protein-1β, granulocyte colony-stimulating factor or granulocyte-macrophage colony-stimulating factor	[23]
In vitro fermentation system using fresh fecal samples from three healthy donors	0.8 g of fucoidans obtained from *Laminaria japonica* for 48 h. Blank samples contained no polysaccharide	Not provided	Decrease in *Enterobacter* spp. while increase in beneficial bacteria as *Lactobacillus* and *Bifidobacterium*	Decrease in pH and increase in lactic acid and SCFA, including acetic and butyic acids	[82]
In vivo trial using 18 male C57BL/6 mice. Six mice received fucoidans extracted *Laminaria japonica*, and six acted as blank group	100 mg/kg/day of fucoidans obtained *L. japonica* for 6 weeks. Control group received saline solution	Fucoidans from *L. japonica* 18.4% sulfate content; 310 KDa molecular weight; 11.2% Man, 7.3% GlcA, 5.2% Glc, 19.3% Gal, 2.9% Xyl, 54.1% Fuc	Increase in the abundance of *Ruminococcaceae*	Decreased in the serum levels of lipopolysaccharide-binding protein	[95]
In vivo trial using six male Wistar rats per group	2% *w*/*w* of laminarins for 2 weeks. Black samples received control diet	Not provided	Increase of *Bacteroides capillosus*, *Clostridium ramosum* y *Parabacteroides distasonis*	Increase organic acids, specially propionate, whereas decreased cecal putrefactive compounds (indole, phenol and H_2_S)	[101]
In vitro fermentation system using fresh fecal samples from three healthy donors and an in vivo trial using 20 Wistar rats	1 g laminarins from *Laminaria digitata* for 24 h. Glucose was used as negative control and FOS as positive control	Not provided	No significant differences were obtained in the in vitro trial for GM composition.	Increase in total SCFA in laminarin-added culture medium than in glucose-added. Laminarins supplementation increased the colon luminal content of mucin, while decreased luminal mucin in jejunum, ileum and caecum in rats	[89]
In vivo trial using 28 female Sprague-Dawley rats	Supplementation with 10% of dried *L. japonica* for 4 weeks. Control rats were fed with basal diet	Not provided	Reduction in Firmicutes to Bacteroidetes ratio and decrease of pathogenic bacteria such as *Clostridium*, *Escherichia* and *Enterobacter*	Increase in total SCFA, and butyric acid. Lower production of acetic acid propionic acids	[90]
In vivo trial using six female BALB/C mice per group	Mice received normal diet, high-fat diet or high-fat diet added with laminarins at 1% *w*/*w* in a high-fat diet ad libitum for 4 weeks. After finishing, highly-fat diet was provided for an additional 2 weeks	Not provided	Decrease in Firmicutes and increase in Bacteroidetes phylum, especially the genus *Bacteroides* in laminarin-added fed mice with respect to controls	Mice fed with laminarin supplementation showed significantly higher CAZyme families in feces	[44]
In vitro fermentation system using fresh fecal samples from three healthy donors	Polysaccharides isolated from *Laminaria digitata* crude or depolymerized (1% *w*/*v* for 48 h). Cellulose was used as negative control and FOS as positive control	Not provided	Increase *Parabacteroides*, *Fibrobacter* and *Lachnospiracease* and decrease in *Streptococcus*, *Ruminococcus* and Peptostreptococcaceae in laminarin-added samples	Increase in SCFA with respect to cellulose-added samples, but similar SCFA content or even lower with respect to FOS-added samples	[91]
In vitro fermentation system with individual bifidobacteria including *B. infantis* JCM 1222; *B. longum* JCM 1217 and *B. adolescentis* JCM 1275	Beta-glucans from *L. digitata* (0.5% *w*/*v* for 24 h) compared to barley β-glucan, Curdlan from *Alcaligenes faecalis*, mushroom sclerotia from *Pleurothus tuber-regium* and inulin	β-Glucan > 95%, protein 3%; monosaccharides: 98% Gluc; 2% Man. 6 kDa as average molecular weight	Increase of all *Bifidobacteria* with respect to initial counts in a similar way of the other beta-glucans assayed	Increase of SCFA, acetic propionic and butyric acids and decrease of pH in a similar way of the other beta-glucans assayed	[92]
In vitro fermentation system in cellular lines using human-enterocyte-like-29-Luc cells	Supplementation with 0.5% *w*/*v* and 0.1% *w*/*v* of sodium alginate and laminarins extracted from *Eisenia bicyclis* for 18 h.	Glu residues with degree of polymerization between 22 and 25 and 5 kDa as average molecular weight	Inhibition of *Salmonella* Typhimurium, *Listeria monocytogenes* or *Vibrio parahaemolyticus* adhesion and invasion	Not provided	[73]
In vivo trial using 24 male C57BL/6J mice	Polysaccharides extracted from *Porphyra haitanensis* (250 mg/kg) for 2 weeks. Control mice received 0.9% normal saline at a dose of 20 mL/kg/day. Positive controls received the same plus combined *Bifidobacterium*, *Lactobacillus* and *Streptococcus thermophilus* tables, 500 mg/kg.	Not provided	Increase of *Prevotellaceae Rikenellaceae* and *Lactobacillus*, while decreased *Lachnoclostridium* or *Lachnospiraceae*	Not provided	[97]
In vivo trial using 16 male C57BL/6 mice fed with a high-fat diet	5% *w*/*w* polysaccharides extracted from *Saccorhiza polyschides* with high-fat diet for 8 months	Not provided	Not provided	Reduced activity of fecal bile salt hydrolase activity and secondary bile acids	[96]
In vivo trial using Syrian golden hamsters	150 mg/kg body weight of *Sargassum confussum* solution once daily for 60 days by intragastric administration	Sulfated oligosaccharide containing galactose, sulfated galactose, sulfated anhydrogalactose and methyl sulfated galactoside	Increased gut bacterial diversity in treated hamsters. Significant increase in *Barnesiella*, *Tannerella*, *Eubacterium* and *Clostridium* XIVa, with significant decrease in *Allobaculum*, *Bacteroides*, and *Clostridium* IV in the *S. confussum*- added group.	*S. confussum* administration significantly reduced the gene expression of JNK1 and JNK2 in hepatic cells and increased expression of IRS1 and PI3K	[15]
In vitro fermentation system using fresh fecal samples from three healthy donors	Extracts from *Sargassum multicum* (1% *w*/*v* for 24 h). FOS was used as positive control and no carbon source was added to the negative control	Not provided	Increase *Bacteroides* and *Prevotella*, and decrease in *Clostridium coccoides* and *Eubacterium rectale*	Increase in SCFA and lactic acid production with respect to negative controls	[88]
In vitro test comparing the growth of *L. plantarum* NCIM 2083 with respect to *Salmonella* Typhimurium MTCC 3224	1% *w*/*v* of enzymatic-extracted polysaccharides from *Sargassum wightii* in MRS broth for 48 h	53.5% fiber, 13.2% protein, 2.3% fat, 28.9% ash. Content of Cel, Fru and Gluc (not specific proportions)	Prebiotic effect by 1.42-fold more growth stimulation of *L. plantarum* than *Salmonella* Typhimurium	Not provided	[86]
In vitro fermentation system using fresh fecal samples from three healthy donors	200 mg polysaccharides extracted from *Sargassum thunberguii* for 48 h	68.3% carbohydrate; 0.3% protein; 3.5% sulfate: Monosaccharide molar ratio: 3.9% arabinose; 6.2% Gal, 3.2% Glc, 15.6% Xyl, 14.8% Man 15.6% GulA, 40.6% GlcA. Average molecular weight 4.8 kDa	Decrease of Firmicutes, while increase of Bacteroidetes and beneficial bacteria such as *Bifidobacterium*, *Roseburia*, *Parasutterella* and *Fusicatenibacter* after 24-h fermentation, and increase of *Faecalibacterium* and *Coprococcus* after 48-h fermentation	Decrease of pH and increase in total SCFA and acetic, propionic, butiric and *n*-valeric acids	[87]
In vivo trial using 28 female Sprague-Dawley rats	Supplementation with 10% of dried *Undaria pinnatifida* and for 4 weeks. Control rats were fed with basal diet	Not provided	Reduction in Firmicutes to Bacteroidetes ratio and decrease of pathogenic bacteria such as *Clostridium*, *Escherichia* and *Enterobacter*	Increase in total SCFA, and butyric acid. Lower production of acetic acid propionic acids	[90]

BCC: Biotec Culture Collection; BCC: British Culture Collection; Cel: Cellobiose; Gal: galactose; GlcA: galacturonic acid; Glc: glucose; GulA: guluronic acid; FOS: fructooligosaccharides; Fru: fructose; Fuc: fucose; JCM: Japan Collection of Microorganism; kDa: kilodaltons; ManA: Mannuronic acid; Man: mannose; MRS: Man, Rogosa and Sharpe; MTCC: Microbial Type Culture Collection and Gene Bank; NCIM: National Centre of Integrative Medicine; SCFA: Short chain fatty acids; Xyl: Xylose.

**Table 2 molecules-25-01004-t002:** Prebiotic effect of different species of red seaweed.

Type of Study	Seaweed and Dosage	Polysaccharides Characterization	Significant Changes in Gut Microbiota	Significant Changes in Metabolites	Reference
In vitro test comparing the growth of *Lactobacillus plantarum* NCIM 2083 with respect to *Salmonella* Typhimurium MTCC 3224	1% *w*/*v* of enzymatic-extracted polysaccharides from *Acanthopora spicifera* in MRS broth for 48 h	45.9% fiber, 10.9% protein, 1.6% fat, 39.4% ash. Only Glu was found as monosaccharide	Prebiotic effect by 0.84-fold more growth stimulation of *L. plantarum* than *S. Typhimurium*	Not provided	[86]
In vivo trial using male Sprague-Dawley rats (six per group)	Fed added with 0.5–2.5% (*w*/*w*) whole *Chondrus crispus* for 21 days	Not provided	Increase of *Bifidobacterium brevis* and decrease of pathogens such as *Clostridium septicum* and *Streptococcus pneumoniae*	Increase in total SCFA and acetic, propionic and butiric acids in rats fed with *C. crispus* at 0.5% and 2.5%. Higher concentrations of all SCFA were found in the case of rats fed added 2.5% of *C. crispus* with respect to rats fed added with 0.5% of *C. crispus*	[57]
In vitro test in MRS broth for *Lactobacillus* and *Bifidofacterium* compared to MHB broth for *Staphylococcus aureus* and *Escherichia coli*	0.1–0.5% *w*/*v Eucheuma spinosum* for 24 h	62.1% sugar; 21.4% sulfate; Monosaccharides at molar ratio: 0.01 Man, 0.01 GluA, 1% Gal, 0.09% Xyl, 0.01% Fuc, 0.03% Glu	Increase in beneficial bacteria with better results at 0.1% concentration. No inhibition was detected against pathogens	Not provided	[10]
In vitro test in MRS broth for *Lactobacillus* and *Bifidofacterium* compared to MHB broth for *S. aureus* and *E. coli*	0.1–0.4% *w*/*v Grateloupia filicina* added in culture media for 24 h	41.9% sugar; 20.6% sulfate; Monosaccharides at molar ratio: 0.01 Man, 0.02 GluA, 1% Gal, 0.1% Xyl, 0.05% Fuc, 0.07% Glu	Increase in beneficial bacteria at all concentrations, without significant differences between 0.4% and 0.5%. No inhibition was detected against pathogens	Not provided	[10]
In vivo trial using male Wistar rats (six per group)	Rats were pretreated with 0.5% carboxymethylcelloluse (controls) or 0.5% *w*/*v* of of sulphated polysaccharides from *Gracilaria birdiae*, twice daily for 2 days. After 1 h, naproxen (80 mg/kg) was administered twice a day for 2 days	Molar mass distribution was found to be within 2.6 × 10^6^ and 3.8 × 10^5^ g/mol, while the soluble carbohydrate, protein, and sulfate contents were 85.5%, 2.5%, and 8.4%, respectively	No relevant variation was observed in GM populations	Prevention of naproxen-induced gastrointestinal damage determined by macro- and microscopic findings	[106]
In vitro fermentation system using fresh fecal samples from four healthy donors	100 mg of sulphated polysaccharides obtained from *Gracilaria rubra* for 24 h. Basal nutrient medium was used for control negative group and FOS was used for control positive group	Average molecular weight 923.3 kDa, sugar content 0.11%	Increase of Bacteroidetes, *Bacteroidaceae*, *Prevotellaceae*, *Ruminococcaceae* and propionic acid, while decrease *Fusobacteriaceae* and *Lachnospiraceae*. At genus level, increase of *Bacteroides*, *Prevotella* and *Phascolarctobacterium*	Increase in total SCFA and acetic, propionic and isobutyric acids	[105]
In vitro fermentation system using fresh fecal samples from three healthy donors	1% *w*/*v* low molecular weight polysaccharide derivatives extracted from *Gracilaria* spp. for 24 h. Inulin was used as positive control and cellulose as negative control	Average molecular weight 143.8 kDa	No significant changes in GM	Increase in total SCFA, acetic and propionic acids	[94]
In vitro fermentation system using fresh fecal samples from three healthy donors	1% *w*/*v* low molecular weight polysaccharide derivatives extracted from *Gelidium sesquipidale* for 24 h. Inulin was used as positive control and cellulose as negative control	Average molecular weight 20.1 kDa and 6.5 kDa, respectively	Only *G. sesquipidale* of 6.5 kDa significantly increased *Bifidobaterium* counts	Both *G. sesquipidale* extracts (20.1 kDa and 6.5 kDa molecular weight) significantly increased total SCFA, acetic and propionic acids	[94]
In vitro fermentation system	1% *w*/*v Kappaphycus alvarezii* for 24 h	Not provided	Increase in *Bifidobacterium*, decrease in *Clostridium coccoides* and *Eubacterium rectale*	Increase in SCFA	[107]
In vitro fermentation system using fresh fecal samples from threehealthy donors	Extracts from *Osmundea pinnatifida* (1% *w*/*v* for 24 h). FOS was used as positive control and no carbon source was added to the negative control	Not provided	Increase in *Bifidobaterium* counts	Increase in SCFA, acetic and propionic acids	[88]

Gal: galactose; Glc: glucose; GulA: guluronic acid; FOS: fructooligosaccharides; Fuc: fucose; GM. Gut microbiota; kDa: kilodaltons; Man: mannose; MRS: Man, Rogosa and Sharpe; MTCC: Microbial Type Culture Collection and Gene Bank; NCIM: National Centre of Integrative Medicine; SCFA: Short chain fatty acids; Xyl: Xylose,

**Table 3 molecules-25-01004-t003:** Prebiotic effect of different species of green seaweed.

Type of Study	Seaweed, Dosage and Time of Exposure	Polysaccharides Characterization	Significant Changes in Gut Microbiota	Significant Changes in Metabolites	Reference
In vivo trial using 36 C57BL/6J mice, 18 males and 18 females in different trials	*Enteromorpha clathrata*, 100 mg/kg/day or 50 mg mg/kg/day for 4 weeks	Molecular weight 11.67 kDa; 14.7% sulfate content. Monosaccharide composition: 1.0% Man, 49.7% Rha, 10.8% GlcA; 29.9% Glc; 1.3% Gal; 7.2% Xyl	Increase of *Akkermansia muciniphila*, *Bifidobacterium* spp., and *Lactobacillus* spp. *E. clathrata* supplementation induced much less alteration in the composition of female GM than in male GM	*E. clathrata* supplementation decreased lipopolysaccharide-binding protein in female mice but not in male mice	[38]
In vitro test comparing the growth of *Lactobacillus plantarum* NCIM 2083 with respect to *Salmonella* Typhimurium MTCC 3224	1% *w*/*v* of enzymatic-extracted polysaccharides from *Enteromorpha compressa* in MRS broth for 48 h	60.6% fiber, 16.9% protein, 1.2% fat, 25.4% ash. Monosaccharide content included cellobiose, fructose, glucose and maltose	Prebiotic effect by 1.44-fold more growth stimulation of *Lactobacillus* *plantarum* than *Salmonella* Typhimurium	Not provided	[86]
In vivo trial using 24 male C57BL/6J mice	Polysaccharides extracted from *Ulva prolifera* (250 mg/kg) for 2 weeks. Control mice received 0.9% normal saline at a dose of 20 mL/kg/day. Positive controls received the same plus combined *Bifidobacterium*, *Lactobacillus* and *Streptococcus thermophilus* tables, 500 mg/kg.	Not provided	Polysaccharides supplementation decreased Tenericutes, and Cyanobacteria. At genus level, decreased *Lachnospiraceae*, *Lactobacillus*, *Mollicutes* and *Mucispirillum*, while increased *Prevotellaceae* and *Rickenellaceae*	Not provided	[115]
In vitro fermentation system using fresh fecal samples from three healthy donors	0.8 g of fucoidans obtained from *Enteromorpha prolifera* for 48 h. Blank samples contained no polysaccharide	Not provided	Decrease in *Enterobacter* spp. in *E. prolifera* fucoidans-added samples	Not significant changes	[82]
In vivo trial using 24 Kunming female mice	Loperamide at a dosage of 9.6 mg/kg/twice a day via oral gavage for 2 weeks was provided to mice to induce slow-transit constipation in mice. Afterwards, *E. prolifera* and polysaccharides extracted from *E.* *prolifera* added in fed at a 1:5 *w*/*v* ratio was administered for 7 days	Not provided	*E. prolifera* increased bacterial diversity, Bacteroidales, Firmicutes, Actinobacteria, and decreased Bacteroidetes and Proteobacteria. Extracts from *E. prolifera* increased *Prevotellaceae*, Firmicutes, Actinobacteria, and decreased Bacteroidetes and Proteobacteria	Both *E. prolifera* and *E. prolifera* extracts diminished histopathological lesions of inflammatory infiltrations in distal colon. Both *E. prolifera* and *E. prolifera* extracts reduced serum levels of nitric oxide (inhibitory neurotransmitter) and showed laxative effects	[114]
In vivo trial using 24 Kunming male mice	Treated mice were fed with high sucrose/high fat diet for 5 weeks. Next type-2 diabetes was induced by intraperitoneal administration of streptozotocin at 45 mg/kg for 3 days. Diabetic mice were administered with 150 mg/kg *E. prolifera* extracts or its flavonoid-rich fractions less than 3 kDa, respectively, for 4 weeks	Not provided	*E. prolifera* extracts increased the proportion of *Alistipes*, *Lachnospiraceae* and *Odoribacter*, while both extracts reduced the proportion of *Ruminiclostridium* and *Akkermansia* in GM of diabetic mice	Flavonoids from *E. prolifera* reduced blood glucose in mice, reduced mRNA expressions of JNK1/2 gene and increased the expression of PI3K, IRS1 and AKT genes in diabetic mice	[22]

Gal: galactose; GlcA: glucuronic acid; Glc: glucose; GM: gut microbiota; kDa: kilodaltons; Man: mannose; MRS: man Rogosa and Sharpe; MTCC: Microbial Type Culture Collection and Gene Bank; NCIM: National Centre of Integrative Medicine; Rha: rhamnose; SCFA: Short chain fatty acids; Xyl: Xylose.

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
