# Peer review of "Potential Use of Marine Seaweeds as Prebiotics: A Review"

_molecules, 2020, doi:10.3390/molecules25041004_

Round 1
Reviewer 1 Report
In the manuscript submitted to Molecules (683160) authors studies the Marine seaweeds and its potential use of prebiotics and effects on human health. This reviewer suggest the publication in Molecules only after major revision.
Theme is interesting as a review, but needs more cites and more comments and comparison.
As an article, the work fails because any new method is showed. No optimizacion, no validation and no application are whowed.
Author Response
The authors would like to thank the editor for giving us the opportunity to revise our manuscript, and the reviewers for their thoughtful and constructive comments, who have made a significant contribution to improving the new version of the manuscript. The manuscript has been revised thoroughly according to comments and suggestions. Changes to the original manuscript have been highlighted in red. An itemized point-by point response to the reviewers’ comments is presented below.
With respect to the comments from the Reviewer 1:
With respect to the comments about more cites and more comments and comparison are required:
According with the suggestion from the Reviewer, in the revised version of the manuscript is were extended the number of references with the inclusion of more than 20 new references. These new references were accompanied with several new comments and comparisons in the main text, which gained more than 3000 new words in the new version of the manuscript.
In concrete, the new references introduced in the revised version of the manuscript were the following:
Cardoso, S.M.; Pereira, O.R.; Seca, A.M.L.; Pinto, D.C.G.A.; Silva, A.M.S. Seaweeds as preventive agents for cardiovascular diseases: From nutrients to functional foods. Drugs 2015, 13, 6838-6865. Kumar, S.A.; Magnusson, M.; Ward, L.C.; Paul, N.A.; Brown, L. Seaweed supplements normalize metabolic, cardiovascular and liver responses in high-carbohydrate, high-fat fed rats. Drugs 2015, 13, 788-805. Seca, A.M.L.; Pinto, D.C.G.A. Overview of the antihypertensive and anti-obesity effects of secondary metabolites from seaweeds. Drugs 2018, 16, 237. Wan-Loy, C.; Siew-Moi, P. Marine algae as a potential source for anti-obesity agents. Drugs 2016, 14, 222. Yang, C.F.; Lai, S.S.; Chen, Y.H.; Liu, D.; Liu, B.; Ai, C.; Wan, W.Z.; Gao, L.Y.; Chen, X.H.; Zhao, C. Anti-diabetic effect of oligosaccharides from seaweed Sargassum confusum via JNK-IRS1/PI3K signaling pathways and regulation of gut microbiota. Food Chem. Toxicol. 2019, 131, 110562. Zao, C.; Yang, C.; Liu, B.; Lin, L.; Sarker, S.D.; Nahar, L.; Yu, H.; Cao, H.; Xiao, J. Bioactive compounds from marine macroalgae and their hypoglycemic benefits. Trends Food. Sci. Technol. 2018, 72, 1-12. Wang, H.M.D.; Li, X.C.; Lee, D.J.; Chang, J.S. Potential biomedical applications of marine algae. Technol. 2017, 244, 1407-1415. Xue, M.; Ji, X.; Liang, H.; Liu, Y.; Wang, B.; Sun, L.; Li, W. The effect of fucoidan on intestinal flora and intestinal barrier function in rats with breast cancer. Food Funct. 2018, 9, 1214-1223. Shin, T.; Ahn, M.; Hyun, J.W.; Kim, S.H.; Moon, C. Antioxidant marine algae phlorotannins and radioprotection: a review of experimental evidence. Histochem. 2014, 116(5), 669-674. Chater, P.I.; Wilcox, M.; Cherry, P.; Herford, A.; Mustar, S.; Wheater, H.; Brownlee, I.; Seal, C.; Pearson, J. Inhibitory activity of extracts of Hebridean brown seaweeds on lipase activity. Appl. Phycol. 2016, 28, 1303-1313. You, L.; Gong, Y.; Li, L.; Hu, X.; Brennan, C.; Kulikouskaya, V. Beneficial effects of three brown seaweed polysaccharides on gut microbiota and their structural characteristics: An overview. J. Food Sci. Tech. 2019, doi:10.1111/ijfs.14408. Wang, H.M.D.; Li, X.C.; Lee, D.J.; Chang, J.S. Potential biomedical applications of marine algae. Technol. 2017, 244, 1407-1415. Collins, K.G.; Fitzgerald, G.F.; Stanton, C.; Ross, R.P. Looking beyond the terrestrial: The potential of seaweed derived bioactives to treat non-communicable diseases. Drugs 2016, 14, 60. Fehlbaum, S.; Prudence, K.; Kieboom, J.; Heerikhuisen, M.; van den Broek, T.; Schuren, F.H.J.; Steinert, R.E.; Raederstorff, D. In vitro fermentation of selected prebiotics and their effects on the composition and activity of the adult gut microbiota. J. Mol. Sci. 2018, 19, 3097. Li, P.; Ying, J.; Chang, Q.; Zhu, W.; Yang, G.; Xu, T. ; Yi, H.; Pan, R.; Zhang, E.; Zeng, X.; Yan, C.; Bao, Q.; Li, S. Effects of phycoerythrin from Gracilaria lemaneiformis in proliferation and apoptosis of SW480 cells. Rep. 2016, 36, 3536-3544. Rosa, G.P.; Tavares, W.R.; Sousa, P.M.C.; Pagès, A.K.; Seca, A.M.L.; Pinto, D.C.G.A. (2019). Seaweed secondary metabolites with beneficial health effects: An overview of successes in in vivo studies and clinical trials. Drugs 2019, 18, 8. Girard, L.; Birse, K.; Holm, J.B.; Gajer, P.; Humphrys, M.S.; Garber, D.; Guenthner, P.; Nël-Romas, L.; Abou, M.; McCorrister, S. Westmacott, G.; Wang, L.; Rohan, L.C.; Matoba, N.; McNicholl, J.; Palmer, K.E.; Ravel, J.; Burgener, A.D. Impact of the griffithsin anti-HIV microbiocide and placebo gels of the rectal mucose proteome and microbiome in non-human primates. Rep. 2018, 8, 8059. Fitzgerald, C., Aluko, R.E.; Hossain, M.; Rai, D.K.; Hayes, M. Potential of a renin inhibitory peptide from the red seaweed Palmaria palmata as a functional food ingredient following confirmation and characterization of a hypotensive effect in spontaneously hypertensive rats. Agric. Food Chem. 2014, 62, 8352-8356. Sato, M.; Hosokawa, T.; Yamaguchi, T.; nakano, T.; Muramoto, K.; Kahara, T.; Funayama, K.; Kobayashi, A.; Nakano, T. Angiotensin I-Conerting enzyme inhibitory peptides derived from Wakane (Undaria pinnatifida) and their antihypertensive effect in spontaneously hypertensive rats. Agric. Food Chem. 2002, 50, 6245-6252. Suárez, Y.; González, L.; Cuadrado, A., Berciano, M.; Lafarga, M.; Muñoz, A. Kahalalide F, a new marine-derived compound, induces oncosis in human prostate and breast cancer cells. Cancer Ther. 2003, 2, 863-872.
With respect to the comments about any new method is showed:
The Reviewer is right in the sense that no new analytical methods were developed to address this manuscript. However, please consider that this is a review manuscript and consequently their main aim is not to develop new analytical methods but provide a compilation of the latest published scientific evidence on the influence of seaweed intake on the human intestinal microbiota. Perhaps this is not the most habitual type of manuscript published in Molecules, but the theme of the monographic issue to which it is addressed is special. Thus, in the special issue guidelines published in the Molecules website is was stated than review articles are welcomed for this special issue, and “prebiotics” is the second keyword listened in the special issue guidelines
Reviewer 2 Report
Comments
There is an increasing research interests in seaweeds as they are rich sources of bioactive compounds and are sustainable crops. The review may contribute to the field. However, the manuscript included mostly in vitro studies. The concept of seaweeds serving as potential prebiotics, or how the changes of microbial compositions by PS from seaweeds benefit host health were not elaborated and comprehensively reviewed. The reviews basically were piling conclusions of in vitro studies. No important details such as study design, method, observations, suggested mechanisms and statistical significance were described. In vitro studies, even fermentation with human fecal inoculum, are screening tests with limitations. Following in vitro screening, the preclinical studies on animals, or human interventions would be more relevant.
Some specific comments:
In the Introduction, author may provide epidemiological evidence that intake of seaweeds indeed improves human health or decrease certain disease risks; Much of the introduction on gut microbiome (lines 62-106) can be described in details in section 4. Not sure how the tables are organized. Based on table 1-3 titles, they can be reorganized according to the species. Another suggestion is to move ref column to the right. Are the changes (all tables, and in texts as well) in microbial composition and SCFAs statistically significant? If not, data should not be included in review. In all 3 tables, in addition to the change in microbial composition and SCFAs, more details such as characterization of the extract/PS, change in metabolites, biomarkers and metabolic consequence should be included.

Author Response
The authors would like to thank the editor for giving us the opportunity to revise our manuscript, and the reviewers for their thoughtful and constructive comments, who have made a significant contribution to improving the new version of the manuscript. The manuscript has been revised thoroughly according to comments and suggestions. Changes to the original manuscript have been highlighted in red. An itemized point-by point response to the reviewers’ comments is presented below.
With respect to the comments from the Reviewer 2:
With respect to the comments about how the changes of microbial compositions by PS from seaweeds benefit host health were not elaborated and comprehensively reviewed:
According to the suggestions from the Reviewer, in the revised version of the manuscript is was included newly information to complement in an important way the information provided in the original version of the manuscript about how seaweeds can benefit host health. Additionally to other phrases and paragraphs introduced in the main text, in the introduction sections it was added the following paragraph:
“Regarding specific benefits on human health, seaweeds have demonstrated to exert preventive effects against several non-transmissible diseases such as cardiovascular diseases [11,12], antihypertensive [13], anti-obesity effects [14] and anti-diabetic effects [15,16], anti-cancer [17,18] or antioxidant activities [19].
Regarding cardiovascular diseases, there are large risk factors contributing that overlap and intertwine, overall contributing to the onset and growth of the disease [11]. Between them, it can be cited a cascade of mechanisms including vascular inflammation, oxidative stress, hypercoagulability and activation of the sympathetic and renin-angiotensin systems. Although in some cases the exact mechanisms through seaweeds can prevent cardiovascular diseases are not always fully understood, it was demonstrated that seaweed consumption can prevent cardiovascular diseases [11,12]. Functional oligosaccharides from seaweeds are associated with a variety of biological processes linked to hypoglycaemic and hypolipidaemic activities, although the concrete mechanisms have not been well studied [15]. With respect to hypertension, various compounds from seaweeds, such as protein-derived bioactive peptides and phlorotannins, can prevent hypertension by inhibition of angiotensin-I converting enzyme activity [13].
The potential anti-obesity activity derived from seaweeds consumption may include a large variety of mechanisms, and involve alteration in lipid metabolism, suppression of inflammation, suppression of adipocyte differentiation and delay in gastric emptying [14]. Between then, an important anti-obesity activity of seaweeds is the inhibition of peroxisome proliferator-activated receptor γ (PPARγ) expression and activation of the adenosine monophosphate-activated protein kinase (AMPK) phosphorylation [13]. Other important anti-obesity mechanism of seaweeds is related with inhibition of lipases, especially pancreatic lipase, that is one of the main therapeutic targets of anti-obesity drugs [14] and was recently demonstrated for various seaweeds species [20]. Additionally, it was reported anti-obesity mechanisms related with concrete seaweed components, such as phlorotannins, that targets the inhibition of adipocyte differentiation or Fucosterol, that decrease the expression of the adipocyte marker proteins PPARγ and CCAAT/enhancer-binding protein alpha. Additionally, seaweeds can prevent obesity by means of the modification of relative quantities of phyla in gut microbiota (GM), and polysaccharides from seaweeds can reduce obesity also by repairing intestinal barrier reducing inflammation [21].
Although abundant signaling pathways have been found to be involved in the process of glucose metabolism, anti-diabetic effects [15,16]. From them, the IRS1/PI3K/JNK/AKT/GLUT4 pathways are important mechanisms in insulin signal transduction. Some seaweed components showed in animal models to ameliorate the hepatic insulin resistance by regulating the cited signaling pathways [15,16,22]. In addition, seaweeds significantly increased the abundance and diversity of gut microbiota in animal models and showed the ability to increase the population of beneficial microbiota and maintain the homeostasis of GM [15,22].
With respect to antioxidant activity of seaweeds, this protective affects falls mainly on phlorotannins, secondary metabolites whose exert great antioxidant via the scavenging of reactive oxygen species [19]. Secondary metabolites of seaweeds are also responsible of anti-cancer activity, whose included different mechanisms such as repairing the intestinal barrier by intensifying the expression of the tight junction proteins via increasing the phosphorylation of MAPK and ERKT/2 genes [18], activating the caspases cascades. Other potential mechanisms are reducing the expression of cyclin-dependent kinases and matrix metalloprotease family [17] and including decreased levels of pro-apoptotic metabolic signals [19].”
Additionally, more information about how seaweeds can enhance human health was cited in the section 3, related to secondary metabolites. In concrete:
“Other important secondary metabolites contained in seaweeds and responsible of important beneficial effect in human health are peptides, such as lectins [48]. Lectins primarily show antiviral, antibacterial, and antifungal activities. Specially one type of lectin, griffithsin, showed important antiviral activity and is nowadays considered a promise antiviral agent, with great potential concerning the prevention of sexually transmitted infections [48], including HIV [64]. Other important peptides are renin inhibitor tridecapeptide [65] and dipeptide [66] which demonstrated hypotensive effect dipeptide. Phycoerythrin [43] and kahalalide F [67] are other important peptidic compounds isolated from seaweeds than showed antitumor effect.”
With respect to the comments about the manuscript includes basically conclusions of in vitro studies:
In the original version of the manuscript, in fact it was included more works that employed in vitro methods (20) that preclinical studies in experimental animals (15). However, it should be considered that our aim was to include the works in whose both seaweeds of polysaccharides extracted from known seaweeds were included, and most of the published works were carried out using in vitro models. In the revised version of the manuscript, we included a newly published article carried out using animal models:
Yang, C.F.; Lai, S.S.; Chen, Y.H.; Liu, D.; Liu, B.; Ai, C.; Wan, W.Z.; Gao, L.Y.; Chen, X.H.; Zhao, C. Anti-diabetic effect of oligosaccharides from seaweed Sargassum confusum via JNK-IRS1/PI3K signaling pathways and regulation of gut microbiota. Food Chem. Toxicol. 2019, 131, 110562.
Other published works using livestock as models are more oriented to study the effects of seaweeds in animal production and welfare and does not are extrapolated to human. We cited this concept in the revised version of the manuscript. It was included in the revised version of the manuscript the following phrase:
“Furthermore, although several works investigated the effects of seaweeds in livestock gut microbiota, they are more oriented to study the effects of seaweeds in animal production and welfare. An elegant review was recently published where they can be checked [32].”
In section 4, it was added a paragraph remarking the limitations of in vitro studies and the need to perform assays in humans:
“Clinical studies investigating prebiotic effects have some disadvantages with respect to ethical constraints, as well as limited sampling possibilities from the colon and limited measurements of in situ SCFA production. These concerns are commonly avoided by applying an in vivo approach [41], that are the most common in the investigation of seaweed effects on human GM, as is described below. Contrariwise, in vitro studies show important limitations because only represent the first step of a long process, and the results observed in vitro can be magnified, diminished, or totally different in a more complex and integrated system [48]. An additional limitation is that, due to the short fermentation time in in vitro studies, they fails to capture the complete picture of cross-feeding interactions between gut microbes, and which may not fully correlate with the long-term effects of seaweed compounds on GM [41].”
Additionally, considering the Reviewer is right in that this study does not show sufficient evidence from human research, the title has been changed to remove the mention of human health. Thus, the new title is:
“Marine seaweeds: Potential use of prebiotics. A review.”
With respect to the comments about no important details such as study design, method, observations, suggested mechanisms and statistical significance were described:
In the revised version of the manuscript we modified the Tables formats to include more details about the type of study, differentiating in vitro studies using human feces than other in vitro trails, including the polysaccharides composition were available in articles, and differentiating changes in gut microbiota than changes in metabolites. All the cited results were previously cited by the authors´ articles as statistically significant. With respect to the suggested mechanisms, please note most of the articles cited do not include such data. Therefore, we do not believe that it is ethical for us to include mechanisms responsible for effects that have not been expressly cited by the authors of the different articles.
With respect to the comments about in the Introduction, author may provide epidemiological evidence that intake of seaweeds indeed improves human health or decrease certain disease risks:
According to the suggestions from the Reviewer, in the revised version of the manuscript is was included newly information to complement in an important way the information provided in the original version of the manuscript about how seaweeds can benefit host health. Specifically, it was including information about the protective effects of seaweeds against cardiovascular disease, hypertension, obesity, diabetes, oxidative stress and some cancers. Please read the answer to the first comment to avoid being repetitive.
Additionally, as was cited previously, the title of the manuscript was modified to remove the reference to human health, as we believe that the reviewer is right in his/her comment
With respect to the comments about much of the introduction on gut microbiome (lines 62-106) can be described in details in section 4:
According to the suggestions from the Reviewer, the information contained in lines 62-106 of the original version of the manuscript was translated into section 4 of the revised version of the manuscript. Some parts of the paragraph were reduced to avoid restate concepts already mentioned
With respect to the comments about not sure how the tables are organized. Based on table 1-3 titles, they can be reorganized according to the species. Another suggestion is to move ref column to the right:
According to the suggestions from the Reviewer, the references column was moved to the right. In the original version of the manuscript, prior to formatting according to Molecules´ instructions for authors, the tables were organized alphabetically by surnames. In fact, after formatting the manuscript according to Molecules´ instructions, it seems confusing. According to the suggestions from the Reviewer, in the revised version of the manuscript, works were reorganized according to seaweed species. Thus, some works than included more than one seaweed specie was divided in various rows.
With respect to the comments about are the changes (all tables, and in texts as well) in microbial composition and SCFAs statistically significant? If not, data should not be included review:
All the cited results were previously cited by the authors´ articles as statistically significant. Please note that there were included an important variety of types of tests, but in our opinion describe all the statistical methods used are too long.
With respect to the comments about in all tables, in addition to the change in microbial composition and SCFAs, more details such as characterization of the extract/PS, change in metabolites, biomarkers and metabolic consequence should be included:
In the revised version of the manuscript we modified the Tables formats to include more details about the type of study, differentiating in vitro studies using human feces than other in vitro trails, including the polysaccharides composition were available in articles, and differentiating changes in gut microbiota than changes in metabolites. With respect to the suggested mechanisms, please note most of the articles cited do not include such data. Therefore, we do not believe that it is ethical for us to include mechanisms responsible for effects that have not been expressly cited by the authors of the different articles.
Round 2
Reviewer 1 Report
After changes, manuscript 683160R2, could be published in its present form.
Author Response
We greatly appreciate the constructive comments made by the reviewer that have helped us to improve the initial version of the manuscript
Reviewer 2 Report
Revision comments:
At the end of responses, authors responded to the major criticism by “we do not believe that it is ethical for us to include mechanisms responsible for effects that have not been expressly cited …” In many publications, if read carefully, there are mechanisms or proposed/suggested ones.
There were a number of poorly written sentences in the revision: e.g. “Regarding cardiovascular diseases, there are large risk factors contributing that overlap and intertwine, overall contributing to the onset and growth of the disease [11]”. (line 66-67)
Overall, authors did not make efforts to improve the manuscript significantly. Some responses did not address the comments or addressed incorrectly. Based on these, I would reject the manuscript.
Author Response
With respect to the comments about “At the end of responses, authors responded to the major criticism by “we do not believe that it is ethical for us to include mechanisms responsible for effects that have not been expressly cited …” In many publications, if read carefully, there are mechanisms or proposed/suggested ones. “
We read carefully the manuscripts cited, and in most cases, direct statements of metabolic changes were not included. In the R1 stage, according to the instructions from the Reviewer, we included information about metabolic pathways of seaweeds effects I human health. In example, in the introduction section it was included as metabolic mechanisms:
Page 2: activation of the sympathetic and renin-angiotensin system, inhibition of angiotensin-I converting enzyme activity, alteration in lipid metabolism, suppression of inflammation, suppression of adipocyte differentiation, delay in gastric emptying, nhibition of peroxisome proliferator-activated receptor γ (PPARγ) expression and activation of the adenosine monophosphate-activated protein kinase (AMPK) phosphorylation, inhibition of lipases, especially pancreatic lipase, inhibition of adipocyte differentiation or fucosterol and decrease the expression of the adipocyte marker proteins PPARγ and CCAAT/enhancer-binding protein alpha, repairing the intestinal barrier, reducing inflammation, scavenging of reactive oxygen species, increasing the phosphorylation of MAPK and ERKT/2 genes, activating the caspases cascades, reducing the expression of cyclin-dependent kinases and matrix metalloprotease family.
Page 7: reducing serum inflammatory markers, reducing serum levels of lipopolysaccharide-binding protein, increasing CAZymes, reducing activity of fecal bile salt hydrolase activity, or reducing the expression or diabetes-related genes.
More examples are included along the main text.
In any case, according to the comments from the Reviewer, in the revised version of the manuscript, we included in the Tables the metabolic effects cited by the authors, in the cases in which the authors mention these effects in a specific way. Accordingly, the same metabolic effects were cited in the main text, to make it correspond with what is shown in the Tables.
With respect to the comments about “There were a number of poorly written sentences in the revision: e.g. “Regarding cardiovascular diseases, there are large risk factors contributing that overlap and intertwine, overall contributing to the onset and growth of the disease [11]”. (line 66-67)”
We are consistent that since none of us is a native English speaker, we can make mistakes in grammar or writing style. In order to avoid such errors, we have used MDPI's official English review service in this new round of review, so we believe that this problem has been solved.
With respect to the comments about “Overall, authors did not make efforts to improve the manuscript significantly”
We are sorry that the reviewer feels that way, but we cannot agree with that statement. In the previous phase of review we made a considerable effort to adequately answer all comments, and as proof of this, please note that the revised manuscript with respect to the original version has increased by more than 3000 and 19 new references were included. In fact, we felt that the revised version had been greatly improved thanks to the reviewers' comments and we specifically appreciated that in the first round of revisions. All tables were modified and formatted to the Reviewer´s instructions, including more data about seaweed and polysaccharides compositions, experiments and changes detected in metabolites.
With respect to the comments about “Some responses did not address the comments or addressed incorrectly. Based on these, I would reject the manuscript.”
We are sorry that the reviewer feels that way. With respect to metabolic changes in the references included in Tables, in the revised version of the manuscript we included metabolic changes that were specifically cited by the authors in the original articles.
Regarding the rest of the responses to the comments, we do not know which ones the Reviewer considers inadequate. We understand that they are consistent and have been documented with the inclusion of new information, but of course we are willing to rebut or supplement our responses if the Reviewer specifies which ones, he or she considers inadequate and for what purpose.